# Caries Experience and Increment in Children Attending Kindergartens with an Early Childhood Caries Preventive Program Compared to Basic Prophylaxis Measures—A Retrospective Cohort Study

**DOI:** 10.3390/jcm11102864

**Published:** 2022-05-19

**Authors:** Stefanie Amend, Thea Hartmann, Monika Heinzel-Gutenbrunner, Roland Frankenberger, Norbert Krämer, Julia Winter

**Affiliations:** 1University Medical Center Giessen and Marburg (Campus Giessen), Medical Center for Dentistry, Department of Pediatric Dentistry, Justus-Liebig-University Giessen, Schlangenzahl 14, 35392 Giessen, Germany; norbert.kraemer@dentist.med.uni-giessen.de; 2University Medical Center Giessen and Marburg (Campus Marburg), Medical Center for Dentistry, Department of Operative Dentistry, Endodontics and Pediatric Dentistry, Philipps-University Marburg, Georg-Voigt-Str. 3, 35039 Marburg, Germany; hartmannthea@gmx.de (T.H.); frankbg@med.uni-marburg.de (R.F.); julia.winter@med.uni-marburg.de (J.W.); 3MH Statistik Beratung, Bienenweg 8, 35041 Marburg, Germany; monika.heinzel@mh-statistik.com

**Keywords:** early childhood caries, caries experience, caries increment, preventive program, fluoride varnish

## Abstract

Dental caries constitutes a public health challenge. As preventive strategies are desirable, this retrospective cohort study aimed to assess the caries experience and increment in children attending kindergartens with an early childhood caries (ECC) preventive program (intervention group, IG) compared to basic prophylaxis measures (control group, CG) located in areas of different socioeconomic status (SES) within Marburg (Germany). The long-term caries experience (2009–2019) of these 3–5-year-old kindergarten children was evaluated. For the caries increment, dental records of 2019 were screened for the availability of a minimum of two dental examinations at least 8 months apart. Caries was scored according to the WHO criteria (dmf–t). The data were split by observation period (300–550 and >550 days). Overall, 135 children (Ø 3.7 years) attended IG, and 132 children (Ø 3.6 years) attended CG. After 300–550 days, no significant differences were found between both groups regarding mean caries increment and experience (*p* > 0.05). After >550 days, IG with low SES exhibited a high caries experience. Fluoride varnish applications could not reduce the caries increment compared to CG in the short-term but slightly decreased the long-term caries experience. Comprehensive ECC prevention measures actively involving parents are needed to overcome the caries burden.

## 1. Introduction

Early Childhood Caries (ECC) has become a “public health problem” due to its high prevalence and rapid progression with far-reaching consequences, such as physical [1,2,3], psychological [4], language, and orthodontic problems [5]. The global burden of disease study from 2015 [6,7] showed that dental caries in the primary dentition is the 12th most common disease, with 560 million affected children across all age groups. ECC is considered one of the most common childhood diseases and is a significant public health concern in deprived communities [8]. However, ECC is not limited to children of low socioeconomic status (SES) [9,10]. Despite a general decline in caries, a minority of children, often from a low socioeconomic background, are affected by a severe form of ECC even in high-income countries [11]. Therefore, the impact of dental, medical, social, and economic costs of ECC has increased all over the world.

The etiology of ECC is complex, although the risk factors—sugary diet from baby bottles [12], sugary foods and beverages [13], non–use and unavailability of fluoridated toothpaste [14,15,16], social, cultural, and environmental factors [12,17], hereditary enamel disorders [18], nutritional status of mother and child [19], oral flora [20,21,22], poor oral hygiene and inadequate plaque control [21], breastfeeding beyond 12 months, especially high frequency and/or nighttime breastfeeding [12,22], saliva volume (decreased flow) and components (especially fluctuations in proteins present) [23,24,25]—are well known.

To overcome the worldwide burden of ECC, the implementation of preventive strategies is important, including timely information and the education of caregivers, dental examinations supporting caries prevention by the inclusion of oral hygiene instructions, and nutrition counselling, as well as the use of fluorides [26].

In general, the recommended treatment approaches for the management of ECC range from non– and minimally invasive strategies for caries prevention and arrest—active surveillance, use of fluorides, and sealing—to more invasive treatments—restoration with fillings or preformed crowns up to tooth extraction—with the treatment choice depending on several individual factors [27]. More invasive treatment approaches aim to restore primary teeth with cavitated carious lesions either directly with various restorative materials—different glass–ionomer cements, compomer, and composite resin—or by using preformed metal or zirconia crowns for cavitated multi-surface lesions [28,29,30,31,32]. To delay or even prevent the entry of the restorative cycle, an ECC preventive program was established in kindergartens within socially deprived areas in the district of Marburg–Biedenkopf located in Hesse (Germany). This program has been implemented for over 20 years now and consists of supervised tooth brushing, oral hygiene measures, nutrition units, dental examinations, frequent fluoride varnish applications (Duraphat^®^, CP GABA GmbH, Hamburg, Germany), parental work, and staff training [33]. Dentists from the public dental health service working in the district of Marburg–Biedenkopf have visited kindergartens on a regular basis to perform dental examinations (once a year) and have provided supervised tooth brushing and fluoride varnish applications (twice a year). Children attending six kindergartens with fluoride varnish applications in the city of Marburg were regarded as the intervention group (ECC intensive prevention group) and were compared to children visiting other kindergartens located outside of the socially deprived areas in the city of Marburg, where supervised tooth brushing, oral hygiene measures, nutrition units, and dental examinations without fluoride varnish application were performed once a year (ECC basic prevention group/control group).

Against this backdrop, this retrospective cohort study aimed to evaluate the long-term caries experience (2009–2019) and the caries increment in children attending kindergartens with an ECC preventive program within areas of low SES compared to basic preventive measures conducted in kindergartens located within areas of higher SES in Marburg, Hesse, Germany. In addition, the results of this retrospective cohort study should provide input for the planning of a prospective cohort study.

The null hypothesis was that there would be no difference between the two groups with regard to long-term caries experience (2009–2019) and caries increment.

## 2. Materials and Methods

### 2.1. Study Design

This retrospective cohort study was approved by the local Ethics Committee of the Medical Faculty of Philipps–University Marburg, Germany (Reg. No. 214/21). During the research project, the ethical principles of the World Medical Association Declaration of Helsinki (2018) were followed [34]. Data were collected during the regular dental health screenings, which are prescribed by law (Hessisches Gesetz über den öffentlichen Gesundheitsdienst (HGöGD), § 10 (8) and § 11) [35]. Once collected, the data were anonymized for further processing. Written informed consent was obtained from the children’s caregivers regarding the ECC preventive program with fluoride varnish application.

### 2.2. Eligibility Criteria and Dental Examination

The following eligibility criteria were applied for inclusion in this retrospective cohort study:

That they were healthy children attending kindergartens in the city of Marburg in 2019.

Children with a completely erupted primary dentition at the first dental examination.

The availability of at least two dental examinations at least 8 months apart.

Children with special needs and children with genetically derived disorders of enamel or dentin—amelogenesis imperfecta, dentinogenesis imperfecta—were excluded from further evaluation.

One experienced dentist (TH) working for the public dental health service conducted all dental examinations within the kindergarten setting. The dentist has participated in all previous calibrations for examinations in Hesse as part of the studies of DAJ (Deutsche Arbeitsgemeinschaft Jugendzahnpflege e. V.) [36,37]. Once the teeth were brushed with toothpaste, the children were seated on a chair, leaning their heads back, and the dental examination was performed using plane mouth mirrors and a halogen spotlight (LEDental, PowerLight lite, Germany, color temperature 5500 K, unit of illumination at 35 cm distance: 20,000 Lux). The children were only asked to swallow their saliva. A blunt dental probe was used to remove the remaining biofilm and to judge the quality of restoration margins. The caries experience was recorded according to the World Health Organization (WHO) criteria using the dmf–t index [38], and initial carious lesions were further noted. All of the data that included general information (examination date, child’s age, and fluoride varnish application) were entered in a special software program (JZP).

### 2.3. Data Extraction

Two data sets were extracted. The first set included the long-term data on the caries experience of 3–5-year-old kindergarten children between 2009/10 and 2018/19, who were examined by one experienced dentist (TH) during the dental screenings. The second part assessed the effectiveness of the ECC preventive program (areas of low SES) compared to basic prophylaxis measures (areas of higher SES) in kindergartens in Marburg with regard to the caries increment.

Anonymized data were extracted independently by one dentist (JW) and cross-checked by a second dentist (SA). The following data were entered into Microsoft Office Excel 2019 spreadsheets (Microsoft Corporation, Redmond, WA, USA): date, age of the child, dental examination on tooth level (dmf–t index), number of teeth with initial carious lesions, including caries progression or arrest, caries risk measured by the criteria of the German Working Group for Youth Dental Public Health (Deutsche Arbeitsgemeinschaft Jugendzahnpflege e. V., DAJ) [39] as a yes/no decision, the degree of dental restorative rehabilitation, fluoride varnish application as a yes/no decision, the period of time between first and second dental examination, and an increase in caries experience between the first and second dental examination (Δ dmf–t).

### 2.4. Statistical Analysis

BioStat Pro (AnalystSoft Inc., BioStat version v7, Walnut, CA, USA) and SPSS^®^ (IBM^®^ SPSS^®^ Statistics Version 28 program, Armonk, NY, USA) were used for statistical analysis. For descriptive data analysis, minima, maxima, means, and standard deviations were computed for the intervention group and the control group. A generalized linear model with an ordinal dependent variable was applied to test if there was a negative linear effect of time on the long-term caries experience. The two groups were matched based on the children’s age and the observation period to obtain homogeneous groups. For inductive statistics, the data were further split by the period between the two dental examinations (300–550 days and >550 days). The nonparametric Mann–Whitney U test was calculated to examine differences in the caries experience between the two groups. The Wilcoxon test was used for the comparison of caries experience between the two examinations. In cross tables, Pearson’s chi-square test was computed. The McNemar test was used for paired samples with dichotomous outcomes. The significance level was set at α = 0.05.

## 3. Results

### 3.1. Long–Term Data concerning Caries Experience

Between autumn 2008 and spring 2019, a minimum of 1201 and up to 1526 children were examined per year during the dental screenings in kindergartens performed by the public dental health service (Table 1).

Over the 10–year period, the mean dmf–t value ranged between 0.81 and 0.64. The maximum caries experience of an individual child varied between a dmf–t value of 14 and 19. A skewed distribution of the caries experience was found, with about 20% of the examined children bearing the entire caries burden and having 3.5–4 carious primary teeth (Table 1).

In the generalized linear model with an ordinal dependent variable, the overall group of 3–5-year-old children showed a significant decrease in caries experience over time (*p* = 0.004). A separate analysis of children from kindergartens with and without fluoride varnish application showed that the decrease in caries experience was only significant in the group of children who had fluoride varnish applications (IG; generalized linear model with ordinal dependent variable; Figure 1: children in kindergartens with fluoride varnish application *p* = 0.018 vs. children from kindergartens without fluoride varnish application *p* = 0.149).

### 3.2. Characteristics of the Intervention and Control Group for Caries Increment Analysis

To ensure that there were no significant age differences and discrepancies in the observation periods between the intervention and the control group, the evaluation of the caries increment had to be restricted to children aged three to four years at the first dental examination and to observation periods of 300–550 days and more than 550 days. After the adjustment of the age and observation period for 358 children examined in 2019 who attended kindergartens and had fluoride varnish applications, 135 children (intervention group) fulfilled the inclusion criteria. The children in the intervention group were on average 3.7 ± 0.4 years old. In total, 69 children were male and 66 females. After the adjustment for the control group, 132 data sets out of the 296 examined children without fluoridation in kindergarten could be included. The mean age of the control group was 3.6 ± 0.4 years with a ratio of 1:1 girl to boy.

### 3.3. Examination Period between 300 and 550 Days 

For the intervention group, the 122 children under examination were 3.7 ± 0.4 years old and the mean observation period was 385.5 ± 66 days. The 91 children aged 3.7 ± 0.5 years of the control group were followed up for 363.6 ± 26.2 days. There were no statistically significant differences regarding the age distribution and the observation period between both groups (*p* > 0.05).

There was a significant increase in the dmf–t value for children attending the intervention group between the first and the second dental examination (Wilcoxon test, *p* < 0.001). At the same time, the level of oral rehabilitation improved from an average of 20.3% to 41.7% (Wilcoxon test, *p* = 0.05).

In the control group, the number of teeth with dentin caries changed significantly from dmf–t 0.8 ± 2.1 at the first to 1.3 ± 2.8 at the second dental examination (Wilcoxon test, *p* < 0.001). Despite the increase in primary teeth with carious lesions extending into the dentin, the level of oral rehabilitation changed by around 2.6% over time (Wilcoxon test, *p* = 0.65).

The comparison of the caries increment revealed no statistically significant differences between both groups regarding the dmf–t values at the first (Mann–Whitney U test, *p* = 0.058) and the second examination date (Mann–Whitney U test, *p* = 0.232). While a significant difference was found in the frequency of caries-free children (dmf–t = 0) compared to the number of children with caries (dmf–t > 0) between the intervention and the control group at the first dental examination (Pearson chi-square, *p* = 0.04), this difference was no longer evident at the second dental examination (*p* = 0.39). At the first and the second dental examination, there was no significant difference between the two groups concerning the number of children with caries experience that were at a high caries risk versus those without a high caries risk (Pearson chi-square, *p* > 0.05). In both groups, about half of the initial carious lesions present at the first examination developed into a breakdown of enamel, which existed as such at the second examination or had already been treated (intervention group: 18 initial carious lesions to nine cases of enamel breakdown, control group: 44 to 18).

### 3.4. Examination Period More Than 550 Days

The data of 57 children (3.6 ± 0.4 years) with a mean follow–up of 745.3 ± 95.6 days were evaluated for the intervention group. The control group consisted of 46 children with a mean age of 3.5 ± 0.2 years and an observation period of 717.3 ± 38.6 days between the two dental examinations. Again, neither the age distribution nor the observation period between the two dental examinations showed statistically significant differences (*p* > 0.05).

Within the intervention group, the mean dmf–t value increased significantly from 0.5 ± 1.3 to 1.9 ± 3.0 during the observation period (Wilcoxon test, *p* < 0.001). Associated with the rise in the number of dentin carious lesions in primary teeth, there was an increase in the percentage of mean oral rehabilitation at the second dental examination (50.7 ± 44.6%) compared to the first one (16.7 ± 38.9%; Wilcoxon, *p* = 0.058).

Between the two dental examinations, the mean dmf–t value also increased significantly in the control group from 0.1 ± 0.7 to 0.6 ± 1.6 (Wilcoxon test, *p* = 0.01). For the small number of children with dental treatment needs due to caries, the difference in the oral rehabilitation rate was not of statistical significance when comparing both time points of the dental examination (Wilcoxon test, *p* = 0.32).

When both groups were compared with each other (Table 2), there were statistically significant differences in the mean dmf–t values at the first and at the second dental examination (Mann–Whitney U test, *p* < 0.05). The mean caries increment was 1.3 ± 2.3 in the intervention group and 0.4 ± 1.1 in the control group (Mann–Whitney U test, *p* = 0.03). This trend was also observed for the number of children showing a caries increment; 21 of 57 children in the intervention group and 8 of 46 children in the control group were affected by new carious lesions at the dentin level over time (Pearson chi-square test, *p* = 0.03). The frequency of caries-free children (dmf–t = 0) versus the number of children with caries experience (dmf–t > 0) in the intervention and in the control group differed significantly at the first (Pearson chi-square, *p* = 0.01) and at the second dental examination (Pearson chi-square test, *p* = 0.04). Nevertheless, the comparison of both groups concerning the number of children with caries experience being at high caries risk and those without increased caries risk showed no statistically significant differences at the first and second dental examination (Pearson chi-square test, *p* > 0.05). During the observation period of more than 550 days, the number of children with dental treatment needs due to caries (McNemar test, *p* > 0.05) and the mean level of oral rehabilitation increased in the intervention and control groups (Wilcoxon test, *p* > 0.05).

There was no significant difference in the caries increment of children in the intervention and in the control group depending on the observation period (300–550 days and more than 550 days).

## 4. Discussion

### 4.1. Results of the Evaluation in Context with Other Studies

In 1999, an ECC preventive program with regular fluoride varnish application was implemented in kindergartens located within socially deprived areas of Marburg–Biedenkopf, Hesse, Germany, because a higher caries experience among these children was noticed during dental screenings of the public health service at the beginning of school. Although this program has been running for more than 20 years, its effectiveness has not been assessed yet. Therefore, the aim of this retrospective cohort study was to assess the long-term caries experience and the caries increment in children attending kindergartens with an ECC preventive program compared to basic preventive measures.

The sample size in the long-term data on caries experience is representative since, according to social reporting, about one-third of the children aged up to five years living in Marburg were included [40]. A stagnation or even a reversal of the positive trend of oral health in the primary dentition can be reported in Germany, at least for the last decades [37]. In the present study, the described stagnation is also evident in the 3–5-year-old kindergarten children with basic prophylaxis measures. The mean dmf–t values of 3–5-year-olds with basic prophylaxis measures in kindergartens from 2013/14 to 2015/16 and 2018/19 were in the range of the mean dmf–t values collected for 3-year-olds in a nationwide study in 2016 [37]. Nevertheless, it remains to be noted that ECC is still a problem in the study region; despite the otherwise low prevalence figures for the kindergarten children with basic prevention compared to the survey of the German Working Group for Youth Dental Public Health [37] and data from the European Union [41]. Around 20% of the children shoulder 100% of the caries burden. This unequal distribution of the caries burden is not only evident in primary teeth [42,43,44,45] but also in permanent dentition [46].

Unfortunately, an effective caries preventive program developed for permanent teeth of schoolchildren [47,48] has not necessarily achieved the same positive effects when transferred to primary teeth of kindergarten children because of two reasons:Compared to permanent dentition, caries progresses faster in primary teeth. The period of amelogenesis is shorter in the primary dentition, leading to a very thin enamel layer and a less organized microstructure [49]. As a result, the enamel of primary teeth is more susceptible to acidic demineralization [50,51].Although parental control should still be present, schoolchildren become more and more responsible for their nutritional and dental health care behavior with increasing age and can be motivated to adopt tooth-healthy behavior. At kindergarten age, parents are fully responsible for the nutritional and toothbrushing behavior of their children. Phillips et al. [52] showed an association between unhealthy maternal behavior and the caries experience of the child. Therefore, the mothers/caregivers need to be trained. This training was not within the scope of the ECC preventive program under investigation.

Nevertheless, two small positive effects of the evaluated ECC preventive program should be pointed out:While a relatively constant and low mean dmf–t value was found when comparing the representative long-term data on the caries experience of 3–5-year-old kindergarten children with basic prophylaxis measures, a significant decline in caries was seen in the kindergarten children from socially deprived areas. This decline in caries may possibly be attributed to regular fluoride varnish applications, where children at the age of five received up to six fluoride varnish applications. The gap between children with and without fluoride varnish application in kindergartens has narrowed slightly over 10 years. Therefore, the null hypothesis—there is no difference between the two groups in regards to long-term caries experience (2009–2019)—must be rejected.In both observation periods (300 to 550 and more than 550 days), the mean level of oral rehabilitation improved more in the intervention group than in the control group. This was also shown in an evaluation of an interdisciplinary ECC preventive program in Dormagen, Germany, and was explained by the interprofessional referral system of midwives, pediatricians, social workers, and dentists of the public health service [53]. For the present study, such a referral system existed between the dental public health service and the dentist in dental practice. However, the children were only referred once a year after the dental examination. Both the screening examination (basic prevention) and the examination with fluoride varnish application, as well as the second fluoride varnish application per year (intensive prevention), were written to the parents. Thus, parents whose children attended kindergartens with the ECC preventive program were confronted with the topic of “dentist” once more per year. Possibly, in the case of untreated ECC, the letter appealed to the conscience of the caregivers and led to a treatment appointment.

There are other findings in the present study that need to be regarded with caution. In the observation period of 300 to 550 days, the comparison between the intervention and control group showed no significant difference concerning the caries increment and mean dmf–t values in the first and second dental examination. Therefore, the null hypothesis—there is no difference between the two groups with regard to the caries increment—must be accepted. Usually, significantly higher dmf–t values can be found in children from socially deprived areas compared to children with a higher socioeconomic family background [42,45,54,55]. In this respect, the results for the observation period of 300 to 550 days may be considered a positive effect.

However, only in the intervention group it was possible to keep the mean caries experience at a constant level, having been 1.3 times higher than in the control group. The evidence for ECC prevention with the application of fluoride varnish is moderate to low [56,57,58,59]. Regarding the evaluation of the caries increment, no significant caries-preventive benefit could be demonstrated for biannual fluoride varnish applications; this is in agreement with a study by Agouropoulos et al. [60], who reported that no effects were visible in addition to tooth brushing with a toothpaste containing 1000 ppm fluoride. In the present study, it can be assumed that at least some of the children brushed their teeth regularly with fluoride toothpaste, although it cannot be ruled out that in addition to 1000 ppm fluoride toothpaste, 500 ppm fluoride toothpaste was also used at home. Officially, teeth brushing was practiced in all kindergartens of the present study, with quality ranging from supervised tooth brushing, which has a positive effect on caries prevention [61], to children brushing without supervision. A systematic review by the Cochrane Collaboration published in 2019 [62] showed that the use of fluoride toothpaste has a caries-preventive effect in comparison to tooth brushing with fluoride-free toothpaste. A fluoride dose-related effect on the decayed, missing, and filled surfaces of teeth in children and adolescents was further observed. Whereas the scientific evidence supports the effectiveness of fluoride toothpaste on caries prevention, the effect of toothpastes containing classical fluoride compounds—amine fluoride and sodium fluoride—on the prevention of erosive tooth wear appears to be limited [63]. With regards to tooth brushing, these confounders were not assessed in the present study but should be taken into account in a prospective cohort study. Promising caries prevention concepts can be found in the literature, which start during pregnancy [64], implement interdisciplinary concepts [45,65], and use motivational interviewing to encourage parents to change their behavior [66] toward a tooth-friendly diet and regular dental care with fluoride toothpaste if caries is already present. In order to reach especially the caregivers of children with increased caries risk, an interdisciplinary prevention program should be based on a low-threshold “come structure” in combination with outreach care [45,67,68]. The prevention concept presented in the study does not adequately reach parents, especially those parents who have children with an increased risk of caries.

### 4.2. Strengths and Limitations

The major strength of this retrospective cohort study is that the prevention measures under investigation have been successfully established in the region of Marburg, Hesse, Germany, for many years. The effectiveness of this concept (“Marburg Model”) for schoolchildren has been documented in various studies [47,48,69,70,71]. In 2000, it was recommended by the leading German associations of health insurance funds as a basic prophylaxis program for group prophylaxis. Additionally, the ECC preventive program was implemented in kindergartens in socially deprived areas in the district of Marburg–Biedenkopf in 1999. The ECC preventive program has reached many children attending the participating kindergartens, as more than 90% have received fluoride varnish applications. Moreover, the caries prevalence data have continuously been documented by one experienced dentist (TH) for ten years, ruling out possible bias due to different examiners.

Then again, the limitations of the present study need to be taken into consideration. A substantial limitation lies within the chosen study design of a retrospective study [72,73,74,75,76]. Strict inclusion criteria were applied to minimize the risk of selection bias [72,75]. In general, the focused outcome, which was the caries prevalence and increment in this study, had existed by the time the dental records were screened [72,75]. Thus, the data extraction depended on the amount and the quality of information kept in the dental records [74]. As far as the quality is concerned, only one experienced dentist (TH) performed the dental examinations and recorded the data for maximum homogeneity [72]. Only 12 of 654 dental records had to be excluded because of conflicting information. For time and organizational reasons, caries was recorded based on the WHO criteria [38]. Future studies should include the recording of initial carious lesions to find significant differences between subgroups even in a population with a low prevalence of caries [77]. This may be achieved by using validated and internationally accepted criteria, such as the International Caries Detection and Assessment System (ICDAS) [78]. However, it must be considered that compressed air must be available to dry the teeth and that examining the children’s teeth will be more time-consuming [79]. Both factors are of importance for dental screenings in a kindergarten setting. Furthermore, the amount of available information did not allow for the collection and the inclusion of behavioral parameters, which are paramount for the etiology of ECC [26]. This fact may have led to biased results, and therefore, behavioral parameters should be mandatorily included in a prospective study.

Low socioeconomic status is not a primary factor, but it is a predisposing etiological factor for ECC [80]. For a bias-free investigation of the ECC preventive program, the intervention and the control group should have had the same SES. This was not considered in the present data analysis and represented a methodological weakness. While there were no differences in age distribution, the mean dmf–t values and observation period between the children with low and high SES, significantly more children in the socially-deprived kindergartens had a caries experience at baseline in the observation period of 300 to 550 days than the children with high SES. This unequal situation at the first examination between the two groups is to be regarded as a confounding factor for the interpretation of the caries increment and must be considered for planning a prospective study.

Finally, the lack of follow-up in some cases needs to be mentioned as a possible source of bias [72,73,75], as many dental records had to be screened to obtain a sufficient number of participants with two dental examinations within the relevant time periods. A possible reason may be the absence of children from dental examinations due to the increased susceptibility to illness in this age group [81]. In the data extraction for the caries increment, the sample size was very small for the period of more than 550 days; therefore, the authors conclude that the observation period for a prospective study should be limited to 550 days.

## 5. Perspectives and Conclusions

Based on the results, limitations, and strengths of the study, the following perspectives and conclusions can be drawn:The children with low SES and fluoride varnish application showed positive long-term effects regarding the reduction in caries. However, the decline in caries cannot be explained by the fluoride varnish application alone, as it may have been influenced by confounding factors that were not collected in the study. By implementing the intensive preventive program, the staff of the dental public health service expected a greater decline in caries. Despite the fact that fluoride varnish applications were not capable of preventing carious lesion formation over a short-term period, it would certainly be a valuable add-on to other long-term preventive programs that provide early and low-threshold information [82] and motivation for parents to adopt tooth-healthy dietary and dental hygiene behaviors [83]. Against this background, the well-implemented fluoride varnish application for kindergarten children from socially underprivileged areas should be maintained and supplemented by another interdisciplinary prevention concept involving gynecologists, social workers, pediatricians, and dentists.After the implementation of further ECC prevention elements, the complete concept should be evaluated prospectively with appropriate parent questionnaires to survey home nutrition and dental care behavior, dental visits, sociodemographic data, etc.The interdisciplinary prevention concept described under point one and used in the Dormagen study [45] should be utilized for all children in the region. For the evaluation of the program, it is important that the respective intervention and the control group do not show any differences during the first dental examination regarding age distribution, mean caries experience, the proportion of children with naturally healthy dentition, and SES.Since the likelihood of the reduction in caries is greatest where a large caries experience is found, future studies should include children attending kindergartens in social hotspots. A kindergarten group, used as a historical comparison, could serve as a control group [45]. The children of the control group should not take part in the interdisciplinary program. Better still would be a comparison with a control group from another region without a preventive program and based on matched pairs [47] in terms of age, SES, and caries experience at the first dental examination.For such a prospective study, it would already be a success if kindergarten children with a low SES were at the same level of caries increment as kindergarten children with a higher SES. In the present population, this would mean a 0.3 dmf–t lower increase in caries for an observation period of about 550 days. Under these conditions, a sample size calculation was carried out and showed that a group sample size of 450 (intervention group) and 450 (control group) achieved a 90% power to detect a difference of 0.3 using a one-sided Mann–Whitney U when the significance level (alpha) of the test is 0.05, and the standard deviation is 1.5 in both groups.

## Figures and Tables

**Figure 1 jcm-11-02864-f001:**
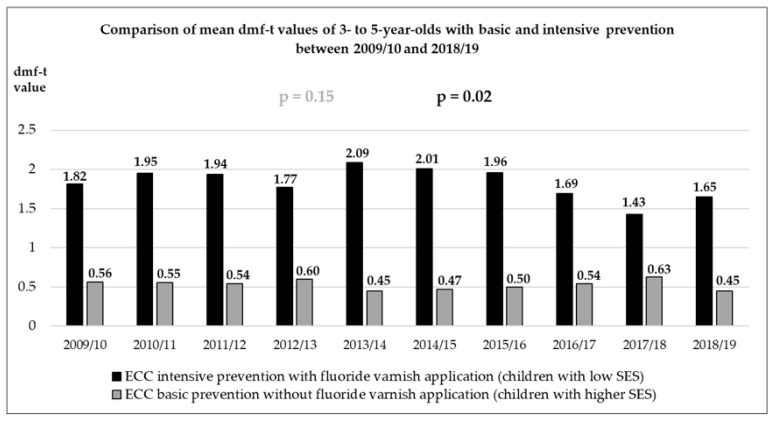
Long-term data from the kindergarten year 2009/10 up to 2018/19 concerning the mean caries experience in 3–5-year-old children attending different kindergartens with and without fluoride varnish application. The p–values given in the figure were calculated with a generalized linear model with ordinal dependent variable to test the linear effect of time within the kindergarten group with ECC preventive program (black) and basic prophylactic measures (grey).

**Table 1 jcm-11-02864-t001:** Long-term data on caries experience of 3–5-year-old kindergarten children (total group TG, control group with ECC basic prevention CG, intervention group with ECC intensive prevention IG).

Year		2009/10	2010/11	2011/12	2012/13	2013/14	2014/15	2015/16	2016/17	2017/18	2018/19
Number of children	TG	1201	1233	1389	1276	1414	1392	1440	1454	1430	1526
	CG	1026	1009	1200	1056	1203	1181	1223	1246	1224	1294
	IG	175	224	189	220	211	211	217	208	206	232
Mean dmf–t value [mean ± SD]	TG	0.75 ± 2.00	0.81 ± 2.12	0.73 ± 2.04	0.80 ± 2.06	0.68 ± 1.95	0.78 ± 2.00	0.72 ± 2.01	0.71 ± 1.98	0.75 ± 2.11	0.64 ± 1.80
Number of children with dmf–t = 0 [n (% of group)]	TG	955 (79.52)	974 (78.99)	1124 (80.92)	1003 (78.61)	1156 (81.75)	1139 (81.82)	1178 (81.81)	1197 (82.32)	1170 (81.82)	1253 (82.11)
	CG	855 (83.33)	856 (84.84)	1011 (84.25)	874 (82.77)	1033 (85.87)	1011 (85.61)	1052 (86.02)	1067 (85.63)	1039 (84.89)	1123 (86.79)
	IG	100 (57.14)	118 (52.68)	113 (59.79)	129 (58.64)	123 (58.29)	128 (60.66)	126 (58.06)	130 (62.50)	131 (63.59)	130 (56.03)
Number of children with dmf–t > 0 [n (% of group)]	TG	246 (20.48)	259 (21.01)	265 (19.08)	273 (21.39)	258 (18.25)	253 (18.18)	262 (18.19)	257 (17.68)	260 (18.18)	273 (17.89)
	CG	171 (16.7)	153 (15.16)	189 (15.75)	182 (17.23)	170 (14.13)	170 (14.39)	171 (13.98)	179 (14.37)	185 (15.11)	171 (13.21)
	IG	75 (42.86)	106 (47.32)	76 (40.21)	91 (41.36)	88 (41.71)	83 (39.34)	91 (41.94)	78 (37.50)	75 (36.41)	102 (43.97)
Mean dmf–t value of children with caries experience total group [mean ± SD (minimum–maximum)]	TG	3.64 ± 2.99 (1–14)	3.85 ± 3.13 (1–15)	3.85 ± 3.13 (1–16)	3.74 ± 2.97 (1–15)	3.80 ± 3.20 (1–18)	3.89 ± 3.12 (1–18)	3.94 ± 3.08 (1–16)	4.00 ± 3.02 (1–17)	4.1 ± 3.27 (1–19)	3.56 ± 2.78 (1–14)
	CG	3.38 ± 2.79 (1–14)	3.65 ± 3.13 (1–15)	3.46 ± 3.05 (1–16)	3.48 ± 2.88 (1–14)	3.18 ± 2.51 (1–12)	3.29 ± 2.75 (1–18)	3.55 ± 2.76 (1–13)	3.77 ± 2.82 (1–16)	4.15 ± 3.49 (1–19)	3.42 ± 2.72 (1–14)
	IG	4.24 ± 3.36 (1–14)	4.12 ± 3.13 (1–14)	4.82 ± 3.12 (1–14)	4.27 ± 3.07 (1–15)	5.00 ± 3.98 (1–18)	5.11 ± 3.44 (1–14)	4.67 ± 3.50 (1–16)	4.51 ± 3.40 (1–17)	3.96 ± 2.67 (1–12)	3.75 ± 2.86 (1–12)

Abbreviations: TG, total group; CG, control group with ECC basic prevention; IG, intervention group with ECC intensive prevention; SD, standard deviation.

**Table 2 jcm-11-02864-t002:** Caries prevalence (n; %), caries experience (dmf–t), caries risk according to DAJ criteria (n; %), level of oral rehabilitation (%), and caries increment (mean; minimum-maximum) among 3–4-year-old children attending the intervention group or the control group with an observation period of 300 to 550 days and of more than 550 days.

Observation Period		300 to 550 Days		More Than 550 Days	
Groups		IG N = 122	CG N = 91	IG N = 57	CG N = 46
dmf–t = 0 and no initial carious lesions present [n (%)]	FDE	80 (65.57)	67 (73.63)	40 (70.18)	42 (91.3)
	SDE	68 (55.74)	50 (54.95)	31 (54.39)	36 (78.26)
dmf–t = 0 and initial carious lesions present [n (%)]	FDE	9 (7.38)	10 (10.99)	5 (8.77)	2 (4.35)
	SDE	4 (3.28)	9 (9.89)	4 (7.08)	1 (2.17)
dmf–t > 0 [n (%)]	FDE	33 (27.05)	14 (15.38)	12 (21.05)	2 (4.35)
	SDE	50 (40.98)	32 (35.16)	22 (38.6)	9 (19.57)
Increased caries risk according to DAJ criteria [n (%)]	FDE	27 (22.13)	13 (14.29)	8 (14.04)	2 (4.35)
	SDE	25 (20.49)	13 (14.29)	8 (14.04)	2 (4.35)
Mean dmf–t [mean ± SD (minimum–maximum)]	FDE	0.96 ± 2.14 (0–12)	0.75 ± 2.14 (0–12)	0.54 ± 1.27 (0–6)	0.13 ± 0.65 (0–4)
	SDE	1.73 ± 2.87 (0–13)	1.34 ± 2.80 (0–14)	1.88 ± 3.04 (0–11)	0.57 ± 1.57 (0–8)
Mean level of oral rehabilitation [%]	FDE	20.32	35.91	16.67	25
	SDE	41.71	38.49	50.67	29.63
Caries increment between FDE and SDE [n (%)]		40 (32.79)	24 (26.37)	21 (36.84)	8 (17.39)
Caries increment [mean ± SD (minimum–maximum)]		0.80 ± 1.48 (0–7)	0.59 ± 1.13 (0–5)	1.33 ± 2.33 (0–11)	0.43 ± 1.05 (0–4)
Caries increment among children with a high caries risk [mean ± SD (minimum–maximum)]		2.52 ± 2.06 (0–7)	2.46 ± 1.51 (0–5)	5.5 ± 3.12 (1–11)	4 ± 0.00 (4–4)

Abbreviations: CG, control group; DAJ, Deutsche Arbeitsgemeinschaft Jugendzahnpflege e. V. [39]; FDE, first dental examination; IG, intervention group; SDE, second dental examination; SD, standard deviation.

## Data Availability

The data presented in this study are available on reasonable request from the corresponding author. The data are not publicly available due to ethical reasons.

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
