# Peer review of "Caries Experience and Increment in Children Attending Kindergartens with an Early Childhood Caries Preventive Program Compared to Basic Prophylaxis Measures—A Retrospective Cohort Study"

_jcm, 2022, doi:10.3390/jcm11102864_

Round 1
Reviewer 1 Report
There are two major defect in M&M of the study groups
1 comparison of the intervention group with F varnish in socially deprived community. low SES. whereas the control group (no F varnish) was high SES (ln 65-67. 88-97, 127-129). The comparison groups should have similar SES rather with the variable as F varnish only.
2 Table 2, data in the FDE which is considered as baseline should be comparable (no statistical significant difference) between the IG and CG (ln 223-225, 255-260). If the baseline data of the 2 groups wa different then it would effect the SDE data and made it uncomparable especially the caries increment between teh 2 periods, FDE and SDE.
Sample size of the data from "Examination period more than 550 days" was very low (57 IG and 46 CG) whereas the calculate sample size should be 450 each group (ln 407-411).
Fig 1 explanation of the 2 groups, ECC preventive program and intensive prevention (blue) and basic prevention (orange) did not reflect the study groups that IG (blue) was from low SES and CG (orange0 from high SES kindergarten children.
Conclusion that F varnish alone was not sufficient to combat ECC from this study (ln 394) might not reflect the discussion (ln 330-331),
Author Response
Dear Reviewer 1,
Thank you very much for your valuable comments. Please find attached our response including the revised text passages.
We hope this manuscript is suitable for publication in your journal and would be very pleased to receive positive feedback.
Yours sincerely, the authors

Reviewer 2 Report
Abstract is well written.
Introduction: from line 58 to line 71: this should be part of the discussion and not of the introduction.
Introduction is well written but should be longer, please focus also on the various treatment available; this reference could be helpful: Ludovichetti FS, Stellini E, Signoriello AG, DI Fiore A, Gracco A, Mazzoleni S. Zirconia vs. stainless steel pediatric crowns: a literature review. Minerva Dent Oral Sci. 2021 Jun;70(3):112-118
Materials and methods: they are really well written and well presented; paragraph 2.2 should be presented in the introduction
Statistical analysis is well performed but it is too long, please describe it more concisely
Figure 1: please, put it in black and white color
Discussion and conclusion are well written and easily readable, please refer also to more reference like: Efficacy of Two Toothpaste in Preventing Tooth Erosive Lesions Associated with Gastroesophageal Reflux Disease
Ludovichetti, F.S., Zambon, G., Cimolai, M., ...Bertolini, R., Mazzoleni, S.Applied Sciences (Switzerland),
Thank you
Author Response
Dear Reviewer 2,
Thank you very much for your valuable comments. Please find attached our response including the revised text passages.
We hope this manuscript is suitable for publication in your journal and would be very pleased to receive positive feedback.
Yours sincerely, the authors

Round 2
Reviewer 1 Report
The authors' response to previous reviewer's comment was mainly added to the discussion stregth and limtation and conclusion of the manusript This did not make the overall paper better in term of M&M, study design and data analysis which affected the result of the study.
Abstract ln 27-29, it was mentioned that intervention group (F varnish) had higher caries experience and caries increment which reverse the concept or hypothesis that F varnish can prevent caries. Mainly this data reflected the low SES in IG had high caries experience and the intervention (F varnish) could not lower the caries increment, in compasiron with the CG (ln 28). Actually the title of the manuscript should cover only the second part of the data (ln 128-131) and the result in Table 2 demonstrated negative finding (Ln 257-258), this was mentioned in the discussion (ln 349-350), but not conformed with the conclusion (ln 428-429).
Some parts of M&M were too much detailed and unnecessaary such as the clinical examination of caries (ln 108-124).
English language should be rechecked such as ln 22-23 swould be better as ln 104. Also "second fluoridation" in ln 331-332 was confused whether fluoride varnish or fluoridation???
There were too many references, some might not necessary besides some were not updated
Author Response
Dear Reviewer 1,
We kindly thank you for the valuable comments. We have revised the manuscript based on your suggestions. Please find enclosed the revised manuscript with changes highlighted in yellow (revision 1) and in turquoise (revision 2).
Yours sincerely,
the authors
